# The Reporting of SDGs by Quality, Environmental, and Occupational Health and Safety-Certified Organizations

**Luis Fonseca** [1,*]  **and Filipe Carvalho** [2]

1   ISEP—P. Porto, School of Engineering and CIDEM R&D, 4249-015 Porto, Portugal
2   IPCA—Polytechnic Institute of Cavado and Ave, College of Technology, 4750-810 Barcelos, Portugal; fcarvalho@ipca.pt
*   Correspondence: lmf@isep.ipp.pt

**Abstract:** Organizations can play a significant role in the advancement of Sustainable Development, and companies with Quality, Environmental, and Occupational Health and Safety (QEOHS)-certified management systems address the three Sustainability Dimensions (economic, environmental, and social). This research aims to map the present level of engagement of those companies in contributing and reporting to the 17 Sustainable Development Goals (SDG) of the United Nations (UN) 2030 Agenda. By publicly disclosing their sustainability reports on their institutional websites, they can, therefore, support this agenda implementation. The content of the company reports that were available by 31 December 2017 in the institutional websites, from a total of 235 Portuguese organizations with QEOHS-certified management systems was analyzed. The results show a moderate reporting of SDGs by those companies, with the top five being SDG 12—Responsible consumption and production (23.8%); SDG 13—Climate action (22.1%); SDG 09—Industry, innovation, and infrastructure (21.3%); SDG 08—Decent work and economic growth (20.0%); and SDG 17—Partnerships for the goals (19.6%). The results of the statistical tests indicate that the communication of SDGs is more prominent in organizations (QEOHS) with the following characteristics: have a high business volume, are members of the United Nations Global Compact Network Portugal, and disclose their sustainability reports on their website. This study can be useful for both managers and decision makers who aim to support organizations in contributing to the Sustainable Development Goals and achieving a better and sustainable future for all.

**Keywords:** Sustainable Development Goals (SDGs); sustainability reporting; quality, environmental and occupational health and safety; certified organizations

## 1. Introduction

Since the United Nations World Commission on Environment and Development released the "Our Common Future" report [1], the concept of sustainable development has been among the most relevant topics worldwide. However, one of the main challenges for sustainability is to operationalize the resolutions of the Brundtland Report, to ensure simultaneous economic development, social development, and environmental protection, and achieve a higher quality of life for all people and protect all living beings and the planet. The 17 Sustainable Development Goals (SDGs) (with 169 other goals) included in the UN's document Transforming our World: The 2030 Agenda for Sustainable Development aim to foster the integration of sustainability into organizations worldwide, addressing current and future stakeholder needs and contributing to the achievement of sustainable development for society at large.

The International Organization for Standardization (ISO) has published over 22,000 International Standards and related documents representing globally recognized guidelines and frameworks based on international collaboration [2]. ISO standards support the economic, environmental, and social pillars of sustainable development and ISO has issued a document outlining how ISO standards contribute to the UN Sustainable Development Goals and how they can help to transform our world as proposed in the United Nations 2030 Agenda [2].

The academic research addressing the implementation of ISO International Standards is a significant area of scientific interest, e.g., Tari et al. [3] and Fonseca et al. [4]. The same is true of the research addressing the incorporation of SDGs, e.g., Topple et al. [5] and Morioka et al. [6]. However, there are still open issues regarding SDG performance measurements, operationalization, and interlinkages [7].

Progress in implementing the United Nations 2030 Agenda should be assessed periodically by each country, involving governments, civil society, business and other stakeholders. In Portugal, the responsibility for overall SDG coordination rests with the Ministry of Foreign Affairs in liaison with the Ministry of Planning and Infrastructure, involving the other Ministries with their SDG-related tasks. The Inter-Ministerial Committee on Foreign Policy (CIPE) acts as the headquarters and forum for inter-ministerial coordination, both for the implementation of SDGs and for the preparation of reports that will support national, regional and global monitoring processes.

Portugal is a European Union (EU) country, member of the Organization for Economic Co-operation and Development (OECD), and both Portugal (e.g., via the EU and the OECD) and the Portuguese (e.g., now, via the present UN Secretary-General; in the past, via, the former EU Commission President) actively engage in international partnerships and institutions. It is expected, therefore, that this research can be replicated in other countries that also want to foster the UN 2030 Agenda, via the monitoring and implementation of SDGs.

This research aims to map the reporting of SDGs by Portuguese organizations that hold simultaneously Quality, Environmental and Occupational Health and Safety certifications, and publicly communicate their sustainability reports on their websites.

Since these International Standards already address (at least partially) the economic, environmental, and social dimensions of SD, this investigation can contribute to gather further knowledge concerning SDG adoption and foster its application by those organizations. By mapping the present level of engagement of those companies in contributing and reporting to the 17 Sustainable Development Goals of the United Nations 2030 Agenda, leading practices and areas for improvement can be identified, creating awareness and supporting decision and policy makers to advance this agenda implementation further. These companies can encourage inclusive and sustainable economic growth, providing employment and decent work for all, advancing sustainable industrialization and fostering innovation, and reducing inequalities, by engaging in favor of SDGs.

This article will proceed as follows. Section 2 presents the literature review. Section 3 presents the methodology. The results are outlined in Section 4, and Section 5 makes a summary of the study discussions, conclusions, limitations, and recommendations for future research.

## 2. Literature Review

### 2.1. Quality, Environmental, and Health and Safety Management Systems

The globalization movement fostered the adoption of voluntary management standards (MS) as a regulatory mechanism to respond to stakeholder concerns related to global organizations and their supply chains [8]. Among the most common voluntary international standards, the international standards management systems for quality (ISO 9001:2015, [9]), environment (ISO 14001:2015; [10]), and occupational health and safety [OHSAS 18001 [11]; ISO 45001:2018, [12]) stand out. These MSs can be audited and certified by independent external certification bodies (CBs). The CBs, by performing a third-party audit, assess whether the applicable MS complies with the reference international standard (e.g., ISO 9001, or ISO 14001, or OHSAS 18001, or ISO 45001) and achieves the intended results [4].

ISO 9001 Quality Management Systems (QMSs) is the most disseminated MS, with over 1 million certified organizations worldwide, covering all activity sectors and organization types and sizes [13]. ISO edited the ISO 9001 series in 1987. In the early years, organizations that adopted and certified their QMSs accordingly to ISO 9001 were mainly focused on the implementation of a documented quality system to support their efforts for globalization [14,15]. In subsequent years, this focus evolved to improve process performance and customer satisfaction, and ultimately to contribute to company survival, as supported by Poksinska, Eklund, Jörn and Jens [16]; Han and Chen [17]; Singh, [18]; Prajogo, [19]; Chatzoglou, Chatzoudes and Kipraios, [20]; Zimon, [21]; Fonseca and Domingues [22]; and Fonseca et al. [23]. There are institutional and economic motivations for the adoption of ISO MSs [24], and a standard must prove its benefits [4]. There is a considerable stream of research that posits that ISO 9001 certification generates both internal and external benefits, such as improved product quality and process performance, cost reductions, and higher quality awareness, leading to enhanced customer satisfaction, a better market image, and a stronger competitive position [3,4,25–27]. However, the successful ISO 9001 QMS implementation and certification is most significant when the motivations are mainly internal (willingness to change and improve) and is related to the way ISO 9001 is interpreted [13]. According to ISO 9001:2015 (Section 0.1. General, [9]), "The adoption of a quality management system is a strategic decision for an organization that can help to improve its overall performance and provide a sound basis for sustainable development initiatives." Research on the contribution of Quality Management (QM) for sustainable development highlighted that QM and integrated management systems are supportive of a sustainable development initiative and Environmental Management System implementation, e.g., Siva et al. [28].

The ISO 9001 International Standard's success contributed to the creation of the Environmental Management Systems (EMS) standards and the subsequent diffusion of ISO 14001 [4], and, consequently, the way business approaches sustainable development [29]. The 1992 Rio de Janeiro summit triggered an increased international emphasis on the development of environmental sustainability and more environmentally-friendly products and services and increased the demand for voluntary EMSs, namely ISO 14001 [10]. This EMS standard is framed in the assumption that better environmental performance can be reached when environmental aspects are systematically identified and managed through pollution prevention, improved environmental performance and compliance with applicable laws, giving a significant contribution to Sustainability [30]. ISO 14001 helps organizations to achieve their environmental and economic targets [31], is a benchmark for companies to operate in an environmentally-friendly manner [32,33], and supports cleaner production practices [34] and business sustainability [35,36]. Organizations adopt ISO 14001 to ensure compliance with specific environmental legislation, improve environmental awareness and performance, reduce waste and emissions, minimize resource consumption, improve its corporate image, minimize risks and respond to stakeholder expectations [37,38]. Among the reported benefits of ISO 14001 implementation are cost-saving benefits due to improved process efficiencies, increased company legitimacy with stakeholders, access to new markets, improved customer satisfaction, minimization of the environmental impacts and the associated risks, compliance with environmental legislation and improvement in the EMS—all contributing to an increased organizational competitiveness [38–41].

In addition to the concerns with quality and environmental management, organizations also need to focus on preventing injuries and health problems related to work activities in workers and to provide a safe and healthy workplace. Before the introduction of ISO 45001 in 2018, OHSAS 18001:2007—Occupational Health and Safety Management was the primary international occupational health and safety management system (OHSMS) adopted worldwide to support organizations eliminating and minimizing occupational health and safety (OHS) risks by taking effective prevention and protection measures. OHSAS 18001 was developed since there was no ISO OHS standard and adopted the Plan-Do-Check-Act (PDCA) approach and a similar structure to ISO 14001 [11]. OHS comprises the conditions and factors that affect or could affect the health and safety of workers, visitors, or any other person present in the workplace, and the implementation and certification

of OHSAS 18001 OHSMS is relevant for many organizations worldwide [42]. The introduction of ISO 45001 [12] is intended to help organizations, independent of their size or sector, to conceive proactive systems to prevent injuries and worsening health problems as a result of occupational activity. ISO 45001 requirements are designed to facilitate the integration of several ISO MSs, such as ISO 9001 QMSs and ISO 14001 EMSs [43]. The potential benefits that can arise from OHSAS 18001:2007/ISO 45001:2018 implementation comprise increased productivity, reduced costs inherent to stoppages and production losses or defects, a reduction in costs with insurance fees and lost workdays and improvement in the quality of services, or the product provided [44]. It also provides a set of relevant elements towards Sustainable Development (SD), namely, with a focus on the social dimension of SD [45,46].

Competitive factors or demands from clients or other relevant stakeholders fostered the adoption of different management models by companies, namely the integration of Quality, Environmental, and Occupational Health and Safety (QEOHS) Management Systems. Scholars all over the world have investigated this topic, and a growing number of organizations implemented integrated MSs to improve and optimize their organizational issues [47–50]. An integrated Management Systems (IMS) interconnects a set of processes through sharing information, human and financial resources, and infrastructure in order to satisfy the needs of different stakeholders [51]. IMS benefits comprise improved efficiency and capacity to meet customer needs; increased employee satisfaction and motivation; better organizational climate with improved communication and knowledge sharing; systematization of procedures, processes, and responsibilities, with less bureaucracy; enhanced organizational image, market competitiveness, and stakeholder relationships [52,53]. ISO strategic decision to adopt common concepts, core text, and high-level structure for ISO 9001:2015, ISO 14001:2015, and ISO 45001:2018 facilitate the harmonization and unity of the IMS and the implementation and integration of other systems. Scholars support the view that the three MSs (QMS, EMS, and OHSMS) respectively match the three Sustainability Dimensions (economic, environmental and social) and mutually reinforce each other [47,48,54–56], and that QEOHS MSs contribute to a successful and balanced SD [53].

## 2.2. Sustainable Development Goals (SDGs)

The concept of Sustainable Development (SD) was introduced in the document entitled "Our Common Future" by the United Nations Commission on Environment and Development's (Brundtland Commission). SD deals with humanity's aspirations of a better life within the limitations imposed by nature, and it was defined as "the development that meets the needs of the present without compromising the ability of future generations to meet their own needs" [1]. Subsequently, Elkington [57] proposed three dimensions for the operationalization of Sustainability (the Triple Bottom Line concept): the simultaneous search for successful economic development (profit), while taking the environment (planet) and social progress and equity (people) into consideration. By 1997, the United Nations Agenda for Development adopted a definition of Sustainability, including the Brundtland definition and the triple bottom line approach: "Development is a multidimensional undertaking to achieve a higher quality of life for all people. Economic development, social development, and environmental protection are interdependent and mutually reinforcing components of sustainable development" [58]. However, for Govindan et al. [59], one of the main challenges for Sustainability is to operationalize the resolutions of the Brundtland Report, and as for Robert, Parris, and Leiserowitz, another way to define sustainable development is how it is measured [60]. Corporate Sustainability (CS), or Corporate Social Responsibility (CSR), has become vital for organizations' long-term success, and encompasses the integration of the triple bottom line of financial profitability, environmental protection and social responsibility into organizations' core purpose and activities [61–63]. Although there is no consensus concerning the concept of CS, and Sustainability, most definitions account for economic, social, and environmental dimensions [64]. Conceptually, they aim for the simultaneous search for successful economic development with social progress and equity and respect for the natural environment, generating value for shareholders, customers, workers, partners, and society in general [65]. Within this

study, CS is used as an "umbrella construct" that could encompass concepts such as SD, CSR, corporate citizenship (CC), business ethics (BE), and triple bottom line [65]. This approach is aligned with the 2012 United Nations Conference on Sustainable Development in Rio: sustainable progress must cover all three dimensions that affect people's life chances (social, economic, and environmental).

Dyllick and Hockerts (p. 131, [64]) proposed as a definition for corporate sustainability "meeting the needs of a firm's direct and indirect stakeholders (such as shareholders, employees, clients, pressure groups, and communities), without compromising its ability to meet the needs of future stakeholders as well." The shared expression of stakeholder needs is currently represented at the global level by the 17 Sustainable Development Goals (SDGs), announced by the 2015 United Nations General Assembly [66]. The proposal to create the SDGs arose in the Rio+20 United Nations Summit of 2012. After a participated process involving multiple stakeholders, the SDGs (successors of the Millennium Development Goals) with a comprehensive set of development goals were agreed on in September 2015, in the United Nations (New York), by 193 countries. The UN's document Transforming our World: The 2030 Agenda for Sustainable Development includes a declaration of the 17 SDGs and 169 other goals, along with monitoring and review measures [67]. The SDGs balance economic, social, and environmental development and comprehend themes such as ending world poverty to undertaking urgent action to combat climate change and its impacts by 2030 [66]. The SDGs aim to inspire the operationalization and integration of Sustainability into organizations worldwide, addressing current and future stakeholder needs, and contributing to the achievement of sustainable development for society at large [68]. Investigations addressing the incorporation of SDGs into the business are a relevant research subject, and there is a stream of scientific works on this topic within the corporate sustainability literature, e.g., Topple et al. [5] and Morioka et al. [6]. The SDGs have already been linked to concepts such as industrial ecology and strategic management to support organizations to positively contribute to the SDGs while building competitive advantage [69]. However, there are still open issues regarding SDG performance measurements, operationalization, and interlinkages [7], hinting for the need for additional research.

### 2.3. The Reporting of SDGs

Sustainability reporting can be defined as the practice of reporting publicly on an organization's economic, environmental, and social sustainability impacts and the reporting of SDGs as the practice of reporting publicly on how an organization addresses the SDGs [66,70]. However, some companies are concerned with receiving negative feedback from the community by disclosing their sustainability programs and impacts [71].

For Lozano [72], sustainability reporting can be an essential driver of an organization's sustainability orientation. Sustainability reports can, therefore, be a driver for organizations to measure, understand, drive, and communicate their efforts towards the SDGs, setting internal goals and managing the transition towards more sustainable development [70]. The United Nations Global Compact is a significant initiative that has been pushing organizations to embrace the commitments to integrate sustainability into its strategy and operations, engaging with society and reporting the ongoing sustainability efforts and progress annually [73]. It is, therefore, expected that the organizations that have joined this initiative are more prominent in sustainability reporting, including the SDGs.

The adoption of an internationally recognized framework, such as the SDGs, for sustainability reporting, and subsequent public disclosure to the relevant stakeholders (e.g., via their institutional websites), can provide a reinforced legitimacy to the organizations that pursue this approach. However, research by Schramade [74] concluded that only a minority of companies currently mention the SDGs in their reports. Rosati and Faria [75] found that only 16% of a total of 408 organizations investigated in 2016 address the SDGs in sustainability reports. They concluded that the reporting of SDGs is related to factors such as larger organization size and a higher level of intangible assets and a higher commitment to sustainability frameworks and external assurance.

## 3. Methodology

### 3.1. Research Hypotheses

The literature review carried out in the previous sections highlighted that the three MSs (QMS, EMS, and OHSMS) respectively match the three Sustainability Dimensions (economic, environmental, and social), and mutually reinforce each other [48,54–56], with QEOHS MSs contributing to successful and balanced SD [53]. Corporate sustainability has become vital for organizations' long-term success [60,61]. It is framed within the economic, environmental and social dimensions [65] and is related to "meeting the needs of a firm's direct and indirect stakeholders without compromising its ability to meet the needs of future stakeholders as well" [64].

The SDGs are a shared expression of stakeholder needs represented at the global level [66]. Also, the research addressing the incorporation of SDGs into business is a relevant topic within the corporate sustainability literature [5,6], and sustainability reporting can be an essential driver of an organization's sustainability orientation [72]. However, research results highlight that only a minority of companies currently mention the SDGs in their reports [74,75].

Larger organizations, or those with a higher commitment to sustainability frameworks and external assurance (e.g., QEOHS certification), show a high level of the reporting of SDGs and public disclosure of their reports [75]. Since QEOHS certification started within the secondary sector (latter expanding to services), SDGs might show the same pattern. The reporting of SDGs might also be more intensive in organizations that are members of the United Nations Global Compact network and, therefore, commit to integrating sustainability into their strategy and operations and annually report progress.

Accordingly, the following research hypotheses are stated as follows:

**Hypothesis 1 (H1):** *The communication of SDGs is more prominent in organizations (QEOHS) with higher business volume;*

**Hypothesis 2 (H2):** *The communication of SDGs is more prominent in organizations (QEOHS) operating in the secondary sector;*

**Hypothesis 3 (H3):** *The communication of SDGs is more prominent in organizations (QEOHS) that are members of the United Nations Global Compact Network Portugal;*

**Hypothesis 4 (H4):** *The communication of SDGs is more prominent in organizations (QEOHS) that disclose their sustainability reports on their website.*

The following section presents the materials and methods that support this investigation. Section 3 provides the results of the study. The final sections present a summary of the study discussions (Section 4) and conclusions, limitations, and recommendations for future research (Section 5).

### 3.2. Data Collection and Sample

In Portugal, by 31 December 2017, there were a total of 698 QEOHS-certified organizations. The research sample (n) consists of 235 organizations—that is, all Portuguese organizations that were certified, within the scope of Quality (ISO 9001), Environment (ISO 14001), Safety and Health at Work (BS OHSAS 18001), as of December 31, 2017, and had made available an institutional website accessible on the Internet, as of July 31 2019, and released their institutional reports at least once, in the last four years. Data were collected between May and July 2019 through exploratory analysis of companies' institutional websites and the latest available versions of computer files in PDF format on the annually published institutional reports were downloaded for subsequent analysis.

### 3.3. Materials and Methods

The content analysis method was adopted as a research method for this investigation, in line with Carvalho et al. [76]. According to Krippendorff [77] (p. 18), "content analysis is a research technique for making replicable and valid inferences from texts (or other meaningful matter) to the contexts of their use." The application of the content analysis technique has been applied in the investigation related to SD organizational disclosure through the corporate website, as supported by Branco and Rodrigues [78], Gill et al. [79], Tagesson et al. [80], Lee et al. [81] and Amran et al. [82].

This investigation adopted the methodology proposed by Bardin [83], which is in line with the works of Gallego ([84], Ho and Taylor [85], Gill et al. [79], Carvalho et al. [76,86], and Carvalho [87]). The definition of the corpus, categories, and units of analysis was made as follows (Table 1 presents the parameters of the content analysis method):

- The documents of analysis (corpus), encompassing the companies' institutional reports (e.g., sustainability reports, integrated reports, environmental reports, management reports, annual reports, governance reports) available on the websites of QEOHS-certified organizations;
- The categories of analysis, in this research, based on the economic environmental and social dimension of SD;
- The units of analysis, as concepts (themes, words, or phrases) that translate SD commitment.

**Table 1.** Parameters of the content analysis method (adapted from Carvalho, [88]).

| Corpus of Analysis (Documents of Analysis) | Categories and Subcategories of Analysis Sustainable Development Goals (SDGs) | Units of Analysis |
|---|---|---|
| Institutional reports disclosed on the institutional website of the organization (i.e., the corpus of analysis). Institutional reports, such as sustainability reports; social responsibility reports; environmental reports; occupational health and safety reports; management reports; accounts and reports; accounts and management reports; financial reports; corporate governance reports; integrated reports) | 01. No poverty<br>02. Zero hunger<br>03. Good health and well-being<br>04. Quality education<br>05. Gender equality<br>06. Clean water and sanitation<br>07. Affordable and clean energy<br>08. Decent work and economic growth<br>09. Industry, innovation, and infrastructure<br>10. Reduced inequalities<br>11. Sustainable cities and communities<br>12. Responsible consumption and production<br>13. Climate action<br>14. Life below water<br>15. Life on land<br>16. Peace, justice and strong institutions<br>17. Partnerships for the goals | Concept (i.e., the theme, word and/or phrase) |

As stated before, in Portugal, by 31 December 2017, there were a total of 698 QEOHS-certified organizations, with 145 (20.8%) included in the 1000 biggest Portuguese companies and 401 (57.4%) belonging to the secondary sector. A total of 59 (8.5%) organizations are members of the UN Global Compact initiative, embracing the commitments of the UN Global Compact to integrate sustainability into their strategy and operations, engaging with society and publicly reporting ongoing sustainability efforts and progress [88] annually, with the aim of supporting the United Nations 2020 Agenda for Sustainable Development [89]. The number (n) of organizations in the sample is 235 (33.7%), representing all organizations that made an institutional website accessible on the internet as of July 31, 2019, available and, additionally, provide at least one institutional report from the last four years.

An exploratory analysis of the institutional website content of the QEOHS-certified Portuguese organizations was carried between May and July 2019 in order to identify and download the latest available versions of computer files in PDF format of the annually disclosed institutional reports. Subsequently, those documents were analyzed individually, and the extracted data were classified and

registered in the research database by applying the technique of content analysis regarding coding and categorization. Data were analyzed dichotomously, assigning to the item the code or value "1—one" (if present), otherwise, assigning to the item the code or value "0—zero" (Haniffa and Cooke [90], p. 405). Software IBM SPSS Statistics®version 22 (International Business Machines—Statistical Package for the Social Sciences) and macro KALPHA version 2007 (macro Krippendorff's $\alpha$) were used to conduct statistical calculations, hypotheses testing and reliability assessment.

The dependent variable Sustainable Development Goals Communication Index (SDGCI) and its mathematical formulation (in line with Carvalho et al. [76], Amran et al. [82], and Haniffa and Cooke [90]) are presented in (Equation (1)):

$$SDGCI_j = \sum_{i=1}^{n_j} \frac{G_{ij}}{M_{ij}} \tag{1}$$

where SDGCI is the Sustainable Development Goals Communication Index, G represents the number of goals that an organization communicates, and M is the maximum number of goals that an organization is expected to communicate. The dependent variables are business volume (BV), activity sector (AS), United Nations Global Compact Network Portugal (UNGC NP) members (UMs), and sustainability reports (SRs). The definition of the dependent variables is presented in Table 2:

**Table 2.** Definition of the independent variables (adapted from Carvalho, [88]).

| Variables | Description (the organization is classified dichotomously (i.e., in binary form) according to … ) |
| --- | --- |
| Business volume (BV) | … the business volume, in euros (€), obtained in 2017. When the business volume (i.e., turnover) of an organization is among the top 1000 in Portugal, the organization is classified as "Greater" (1); otherwise, it is classified as "Other" (0) |
| Activity sector (AS) | … the activity sector. When the activity sector (i.e., economic sector or industrial sector) of an organization is framed on the secondary sector (second sector), the organization is classified as "Second sector" (1); otherwise, it is classified as "Other" (0) |
| UNGC NP members (UM) | … the relationship with the UNGC NP. When the organization belongs to an economic group that assumes a relationship (i.e., member) with the UNGC NP, the organization is classified as "Member" (1); otherwise, it is classified as "No" (0) |
| Sustainability reports (SR) | … the disclosure of the sustainability reports on the institutional website. If the organization has disclosed a sustainability report on their website, the organization is classified as "Disclose" (1); otherwise, it is classified as "No" (0) |

In the investigation, the estimation of the profile of the Portuguese organizations certified in quality, environment, and health and safety (QEOHS), whose reporting and public disclosure on Sustainable Development Goals Communication is prominent (i.e., above average), was based on "logistic regression". For Kleinbaum and Klein [91], logistic regression "is a modelling approach mathematics that can be used to describe the relationship of independent variables with a dichotomous dependent variable" (p. 5). The proposed estimation model is supported by Equation (2), which was based on the mathematical assumptions of binary logistic regression [91,92], and, in turn, the dependent variable and independent variables are all binary (0, 1). Therefore, the following Binary logistic regression model was used to test the research hypotheses (Equation (2)) statistically:

$$logit \; [P(\text{SDGCI}_{(0,\,1)j} = 1 | \text{BV, AS, UM, SR})] = \beta_0 + \beta_1\text{BV}_j + \beta_2\text{AS}_j + \beta_3\text{UM}_j + \beta_4\text{SR}_j + \varepsilon_j \qquad (2)$$

where,

SDGCI (0,1)—Sustainable Development Goals Communication Index (binary)
BV—Business volume
AS—Activity sector
UM—UNGC NP members
SR—Sustainability reports
β—Regression coefficients
ε—Error term
logit—Link function
P—Conditional probability
j—Organization

## 4. Results

### 4.1. Descriptive Analysis

The descriptive analysis of the results highlights that the SDGs that have a higher reporting frequency (SDGs: 12, 13, 9, 8, 17 and 6) are balanced within the three pillars of SD (Economic: ECO; Environmental: ENV; Social: SOC) as shown in Table 3 and Figure 1 below:

**Table 3.** Communication of sustainable development goals (adapted from Carvalho, [88]).

| Sustainable Development Goals (SDGs) | SD DIM | N | % |
|---|---|---|---|
| SDG 01. No poverty | SOC | 24 | 10.2 |
| SDG 02. Zero hunger | SOC | 26 | 11.1 |
| SDG 03. Good health and well-being | SOC | 37 | 15.7 |
| SDG 04. Quality education | SOC | 37 | 15.7 |
| SDG 05. Gender equality | ECO and SOC | 38 | 16.2 |
| SDG 06. Clean water and sanitation | ENV and SOC | 45 | 19.1 |
| SDG 07. Affordable and clean energy | ECO and ENV | 41 | 17.4 |
| SDG 08. Decent work and economic growth | ECO and SOC | 47 | 20.0 |
| SDG 09. Industry, innovation, and infrastructure | ECO | 50 | 21.3 |
| SDG 10. Reduced inequalities | ECO and SOC | 35 | 14.9 |
| SDG 11. Sustainable cities and communities | ENV and SOC | 29 | 12.3 |
| SDG 12. Responsible consumption and production | ECO and SOC | 56 | 23.8 |
| SDG 13. Climate action | ENV | 52 | 22.1 |
| SDG 14. Life below water | ENV | 38 | 16.2 |
| SDG 15. Life on land | ENV | 41 | 17.4 |
| SDG 16. Peace, justice and strong institutions | SOC | 28 | 11.9 |
| SDG 17. Partnerships for the goals | ECO, ENV and SOC | 46 | 19.6 |

Note: SDG, Sustainable Development Goal; Sustainable Development Dimension (ECO—Economic; ENV—Environmental; SOC—Social); N, number; %, percentage.

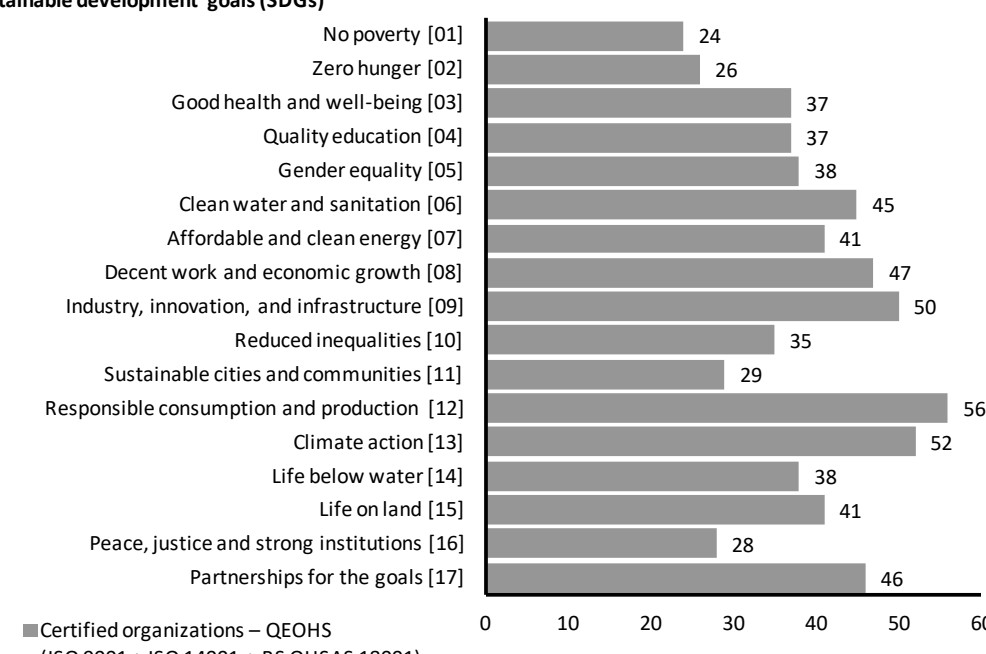

**Figure 1.** Communication of SDG goals in institutional reports (adapted from Carvalho [88]).

*4.2. Univariuate, Bivariate and Multivariate Analysis*

The results obtained from the analysis of the statistical parameters are presented in Table 4. The descriptive characteristics that characterize the continuous dependent variable called SDGCI show a minimum SDGCI of 0.000 and a maximum of 1.000 for 235 organizations—meaning the range of the dependent variable SDGCI was totally filled. The SDGCI mean value equals 0.168, with a standard deviation of 0.306 and a variance of 0.093, hinting the occurrence of high dispersion and variability among the organizations analyzed. Four independent variables (BV, AS, UM, and SR) are a dummy or binary variables (qualitative) and assume the value of 0 or 1 according to their classification category (see Table 5). In terms of statistical dimension, all categories (0 or 1) include at least 45 organizations:

**Table 4.** Statistical results of the characterization of the dependent variable.

| Dependent Variable | *N* | Minimum | Maximum | Sum | Mean | *SD* | Variance |
|---|---|---|---|---|---|---|---|
| Sustainable Development Goals Communication Index (SDGCI) | 235 | 0.000 | 1.000 | 39.412 | 0.168 | 0.306 | 0.093 |

Note: *N*, number; *SD*, standard deviation; SDGCI, Sustainable Development Goals Communication Index.

Concerning the bivariate analysis, relevant differences in the calculated values of the sum and average (dependent variable) by categories 0 and 1 (independent variables) were detected (see Table 5). The statistical assumptions of the normality of the dependent variable and the homogeneity of variances between the categories 0 and 1 were not conclusive. Therefore, to assesses the significance of differences detected, the nonparametric Mann–Whitney U test [93,94] was adopted. The Kolmogorov–Smirnov with Lilliefors correction and the Shapiro–Wilk tests were applied for the study of normality of distribution, and the assessment of the homogeneity of variances was carried with the Levene test. Table 6 presents the results of the Mann–Whitney U test breakdown by research hypothesis.

**Table 5.** Statistical results of the relationship between variables.

| Variables | | | Dependent | | | | | |
| | | | Sustainable Development Goals Communication Index | | | | | |
| **H** | **Independent** | **N** | **Minimum** | **Maximum** | **Sum** | **Mean** | **SD** | **Variance** |
|---|---|---|---|---|---|---|---|---|
| H1 | Business volume | | | | | | | |
| | (0) Other | 141 | 0.000 | 1.000 | 18.824 | 0.134 | 0.288 | 0.083 |
| | (1) Greater | 94 | 0.000 | 1.000 | 20.588 | 0.219 | 0.325 | 0.106 |
| H2 | Activity sector | | | | | | | |
| | (0) Other | 103 | 0.000 | 1.000 | 20.000 | 0.194 | 0.341 | 0.116 |
| | (1) Second sector | 132 | 0.000 | 1.000 | 19.412 | 0.147 | 0.275 | 0.075 |
| H3 | UNGC NP members | | | | | | | |
| | (0) No | 190 | 0.000 | 1.000 | 20.882 | 0.110 | 0.252 | 0.063 |
| | (1) Member | 45 | 0.000 | 1.000 | 18.529 | 0.412 | 0.386 | 0.149 |
| H4 | Sustainability reports | | | | | | | |
| | (0) No | 129 | 0.000 | 1.000 | 11.765 | 0.091 | 0.259 | 0.067 |
| | (1) Disclose | 106 | 0.000 | 1.000 | 27.647 | 0.261 | 0.332 | 0.110 |

Note: H, hypothesis; *N*, number; *SD*, standard deviation.

Since the significance level is 0.05 (confidence level of 95 per cent), the results of the Mann–Whitney U test, presented in Table 6, provide statistical evidence ($p$-value = 0.000) to conclude that there are significant differences $p$-value < 0.05) in the dependent variable SDGCI for categories 0 and 1 of three independent variables (BV, UM, SR). In this sense, the results suggest that individually these three independent variables, according to their category 0 or 1, contribute significantly to a "lower" or "greater" calculated value of the average of the dependent variable by category.

**Table 6.** Statistical results of the Mann–Whitney U test.

| Variables | | | Dependent | | | |
| | | | Sustainable Development Goals Communication Index | | | |
| **H** | **Independent (Categories)** | **N** | **Sum of Ranks** | **Mean of Ranks** | **Mann–WhitneyU Test** | **$p$-Value (One-Tailed)** |
|---|---|---|---|---|---|---|
| H1 | Business volume | | | | | |
| | (0) Other | 141 | 15574.000 | 110.450 | 5563.000 | 0.005 |
| | (1) Greater | 94 | 12156.000 | 129.320 | | |
| H2 | Activity sector | | | | | |
| | (0) Other | 103 | 12414.000 | 120.520 | 6538.000 | 0.263 |
| | (1) Second sector | 132 | 15316.000 | 116.030 | | |
| H3 | UNGC NP members | | | | | |
| | (0) No | 190 | 20555.000 | 108.180 | 2410.000 | 0.000 |
| | (1) Member | 45 | 7175.000 | 159.440 | | |
| H4 | Sustainability reports | | | | | |
| | (0) No | 129 | 13124.000 | 101.740 | 4739.000 | 0.000 |
| | (1) Disclose | 106 | 14606.000 | 137.790 | | |

Note: H, hypothesis; *N*, number; *p*-Value, probability value or significance (one-tailed).

The binary logistic regression model was applied for the multivariate analysis of the mapping of the profile of the certified Portuguese organizations (QEOHS), in which the SDGCI (0, 1) is more prominent.

The assumption of the absence of multicollinearity between the independent variables that set the binary logistic regression model was tested (see Appendix A, Tables A1–A5), and the results suggested the absence of multicollinearity. The statistical results of the binary logistic regression model, which encompass the joint statistical analysis of the four independent explanatory variables, are presented in Table 7.

**Table 7.** Statistical results of the binary logistic regression model.

| H | Independent Variables | $\beta$ | SE | Exp($\beta$) | Wald | *p*-Value |
|---|---|---|---|---|---|---|
| H1 | Business volume | 0.770 | 0.355 | 2.159 | 4.704 | 0.030 |
| H2 | Activity sector | −0.032 | 0.360 | 0.968 | 0.008 | 0.928 |
| H3 | UNGC NP members | 2.003 | 0.407 | 7.413 | 24.270 | 0.000 |
| H4 | Sustainability reports | 1.671 | 0.367 | 5.319 | 20.721 | 0.000 |
| | Constant | −2.638 | 0.396 | 0.071 | 44.335 | 0.000 |
| | Statistical parameters of the binary logistic regression model: | | | | Statistics | *p*-Value |
| | Overall statistics—Chi-square ($\chi^2$) | | | | 57.353 | 0.000 |
| | Overall percentage—Percentage correct (%) | | | | 79.100 | – |
| | Omnibus tests of model coefficients—Chi-square ($\chi^2$) | | | | 59.837 | 0.000 |
| | −2 Log likelihood | | | | 219.228 | – |
| | Cox and Snell—R-square ($R^2$) | | | | 0.225 | – |
| | Nagelkerke—R-square ($R^2$) | | | | 0.323 | – |
| | Hosmer and Lemeshow test—Chi-square ($\chi^2$) | | | | 6.624 | 0.469 |

Note: H, hypothesis; $\beta$, regression coefficient; SE, standard error; Exp($\beta$), exponential regression coefficient; Wald, statistic test; *p*-Value, probability value or significance (two-tailed).

With a significance level of 0.05, the statistical results of the binary logistic regression model, supported by the Wald test, show, with significant statistical evidence (*p*-value < 0.05), that three independent variables (BV, UM, PT and SR) contribute significantly to the values calculated in the category "more prominent" (1) of the dependent variable (SDGCI(0, 1)), when adjusted to the logit function. Since the statistical parameters of the binary logistic regression model present significant statistical evidence, it can be stated that the proposed regression model has a moderate adjustment power.

The results of the statistical tests following the application of the binary logistic regression model are summarized in Table 8.

**Table 8.** Statistical results obtained by the application of hypothesis testing.

| Research Hypotheses Tested with the Binary Logistic Regression Model | | | |
|---|---|---|---|
| H1 | H2 | H3 | H4 |
| Accept | Reject | Accept | Accept |

Note: H, hypothesis.

## 5. Discussion and Conclusions

The United Nations 2030 Agenda for Sustainable Development requires the collaboration of multiple stakeholders for the successful implementation of the 17 Sustainable Development Goals (SDGs). Organizations can play a significant role in the advancement of the Sustainable and its recognized that those with Quality, Environmental, and Occupational Health and Safety (QEOHS)-certified management systems respectively match the three Sustainability Dimensions (economic, environmental, and social). This research aims to map the present engagement level of those companies in addressing and reporting the SDGs and in publicly disclosing their sustainability reports on their institutional websites. The content of companies reports available in the respective websites, by 31 December 2017, of a total of 235 Portuguese organizations with QEOHS-certified management systems, was analyzed. The results show a moderate reporting of SDGs by those companies, with

the top five being SDG 12—Responsible consumption and production (23.8%); SDG 13—Climate action (22.1%); SDG 09—Industry, innovation, and infrastructure (21.3%); SDG 08—Decent work and economic growth (20.0%); and SDG 17—Partnerships for the goals (19.6%). These results are consistent with Schramade [74] and Rosati and Faria [75] conclusions that most companies currently do not mention the SDGs in their reports. Although 23.8% is higher than the 16% found by Rosati and Faria [75], this indicates that there is still considerable room for improvement in this regard.

The results of the statistical tests have pointed out that the communication of SDGs is more prominent in organizations (QEOHS) with higher business volume, which is in line Rosati and Faria [75] claims that larger organizations show a high level of the reporting of SDGs. The results of the hypotheses testing did not support the assumption that organizations of the secondary sector are more prominent in reporting the SDGs. However, concerning the United Nations Global Compact, the reporting of SDGs is indeed higher within members organizations, consistent with the network purposes. Finally, the results also confirm that the organizations that publish sustainability reporting are more prominent in reporting SDGs, supporting Lozano [72] claims that sustainability reporting can be an essential driver of an organization's sustainability orientation. Therefore, larger organizations and those organization members of the United Nations Global Compact can have a significant role in advancing SDG adoption and reporting within their supply chain.

Portugal is a OECD member with a similar pattern to other OECD countries, where small and medium enterprises (SMEs) are the predominant form of enterprise, accounting for approximately 99% of the business fabric, accounting for 70% of jobs and for creating between 50% to 60% of the added value [95]. According to the OECD [96], SMEs have an essential role to play in favor of SDGs, as they can promote inclusive and sustainable economic growth, providing employment and decent work for all, promoting sustainable industrialization and fostering innovation, and reducing inequalities. The OECD [96] specifically highlights the relevance of SMEs for encouraging the adoption of the more economic SDGs, such as SDG 8 and SDG 9 in the OECD. This is supported by the results of this investigation that have identified the following SDGs as being among the most reported SDGs by the Portuguese QEOHS-certified companies: SDG 09—Industry, innovation, and infrastructure (21.3%); SDG 08—Decent work, and economic growth (20.0%).

This research makes a novel contribution by mapping the reporting of SDGs by Portuguese organizations that hold QEOHS certifications simultaneously. There is a considerable stream of research covering QEOHS certification, and the organizations that adopt these International Standards already address (at least partially) the economic, environmental, and social dimensions of SD. By mapping the present level of engagement of those companies in contributing and reporting to the 17 SDGs of the United Nations 2030 Agenda, leading practices and areas for improvement can be identified, creating awareness and supporting decision among both QEOHS-certified organizations' management and policy makers, to further advance this agenda implementation. We can conclude that Portuguese QEOHS-certified companies have the potential to contribute to all the SDGs, and it is up to each company to identify which priority SDGs are based on their environmental, social and governance impacts along the value chain.

Although this research makes a novel contribution to the SDG body of knowledge, particularly within the QEOHS-certified organizations, it suffers from some limitations to be acknowledged when generalizing its findings. First, the sample is restricted to Portuguese organizations with certified QEOHS management systems. Second, the investigation is restricted to the reporting of SDGs in company reports available on websites without evaluating the performance in terms of SDG advancement. Third, other organizational factors such as resources and capabilities and sustainability performance were not investigated. Fourth, it should be assessed if these results and conclusions can be generalized to other EU and OECD countries, as there is research underlining statistical differences in environmental and social performance between developed EU countries and developing EU countries [97]. Moreover, countries must interpret the SDGs according to their national circumstances and levels of development [98]. Fifth, the SDGs' logic implies that there is mutual dependence between

SDGs (drawing analogies with the integration of QEOHS Management System) [98], but this was not subject to this research.

Therefore, future research could be carried out to evaluate the evolution of the reporting of SDGs with time, the relationships between SDGs, and consider other organizations apart from QEOHS-certified organizations, and in other countries. Also, the application of the data mining process, such as the Cross Industry Standard Process for Data Mining (CRISP-DM), could be adopted, allowing the application of text mining methodology on the digital archives of the sampled companies.

This study contributes to the sustainable development and sustainability reporting body of knowledge by mapping the present engagement level of QEOHS-certified Portuguese companies in addressing and reporting the SDGs, and it can be of value for other researchers that want to investigate and contribute to the UN 2030 Agenda. It can be useful for decision and policy makers that aim to support organizations in contributing to the Sustainable Development Goals and the adoption of the UN 2030 Agenda [89]. From a managerial perspective, it highlights that QEOHS-certified organizations that already address the economic, environmental, and social dimensions, due to their management system certification, can be more ambitious and match their strategies and actions with the relevant SDGs and report accordingly. Considering that the reporting of SDG goals is still modest, more pressure from stakeholders to encourage this and more noteworthy companies disseminating the adoption of SDGs within their supply chains in order to achieve a better and sustainable future for all is desired.

**Author Contributions:** Conceptualization: L.F. (40%) and F.C. (60%.); methodology: L.F. (30%) and F.C. (70%); investigation: L.F. (30%) and F.C. (70%.); resources: L.F. (25%) and F.C. (75%.), writing—original draft preparation: L.F. (75%) and F.C. (25%.); writing—review and editing L.F. (75%.) and F.C. (25%); visualization L.F. (50%) and F.C. (50%).

**Funding:** This research received no external funding.

**Acknowledgments:** The author(s) thank Manuel Gilberto Freitas dos Santos and Joaquim José de Almeida Soares Gonçalves (IPCA—Polytechnic Institute of Cavado and Ave), and Paulo Alexandre da Costa Araújo Sampaio and José Pedro Teixeira Domingues (UM - Minho University) for their support in previous research projects that preceded this investigation.

**Conflicts of Interest:** The authors declare no conflict of interest.

## Appendix A

Verification of the statistical treatment assumptions: normality; homogeneity of variance.
- Mann–Whitney U test

**Table A1.** Statistical results of the tests of normality to the dependent variable.

| Research Variables | | | | Tests of Normality | | | | | |
|---|---|---|---|---|---|---|---|---|---|
| | | | | Kolmogorov–Smirnov * | | | Shapiro–Wilk | | |
| Dependent | | | | Statistic | df | p-Value | Statistic | df | p-Value |
| SDG CI | Sustainable Development Goals Communication Index | | | 0.428 | 235 | 0.000 | 0.603 | 235 | 0.000 |
| Independent | | | Category | Statistic | df | p-Value | Statistic | df | p-Value |
| BV | Business volume | 0 | Other | 0.466 | 141 | 0.000 | 0.521 | 141 | 0.000 |
| | | 1 | Greater | 0.367 | 94 | 0.000 | 0.702 | 94 | 0.000 |
| AS | Activity sector | 0 | Other | 0.424 | 103 | 0.000 | 0.609 | 103 | 0.000 |
| | | 1 | Second sector | 0.431 | 132 | 0.000 | 0.600 | 132 | 0.000 |
| UM | UNGC NP members | 0 | No | 0.463 | 190 | 0.000 | 0.501 | 190 | 0.000 |
| | | 1 | Member | 0.257 | 45 | 0.000 | 0.818 | 45 | 0.000 |
| SR | Sustainability reports | 0 | No | 0.498 | 129 | 0.000 | 0.391 | 129 | 0.000 |
| | | 1 | Disclose | 0.331 | 106 | 0.000 | 0.764 | 106 | 0.000 |

Note: (*) Lilliefors significant correction; SDGCI, Sustainable Development Goals Communication Index; BV, business volume; AS, activity sector; UMs, UNGC NP members; SR, sustainability reports; *df*, degrees of freedom; *p*-Value, probability value or significance (two-tailed).

**Table A2.** Statistical results of the test of homogeneity of variance in the relationship of the variables.

| Research Variables | | Dependent | Sustainable Development Goals communication index | | | |
|---|---|---|---|---|---|---|
| | | Statistical Parameters | Tests of Homogeneity of Variance | | | |
| Independent | | | Levene statistic | *df1* | *df2* | *p*-Value |
| BV | Business volume | Based on mean | 6.538 | 1 | 233 | 0.011 |
| | | Based on median | 4.483 | 1 | 233 | 0.035 |
| AS | Activity sector | Based on mean | 6.611 | 1 | 233 | 0.011 |
| | | Based on median | 1.377 | 1 | 233 | 0.242 |
| UM | UNGC NP members | Based on mean | 28.161 | 1 | 233 | 0.000 |
| | | Based on median | 26.969 | 1 | 233 | 0.000 |
| SR | Sustainability reports | Based on mean | 30.469 | 1 | 233 | 0.000 |
| | | Based on median | 19.330 | 1 | 233 | 0.000 |

Note: BV, business volume; AS, activity sector; UMs, UNGC NP members; SR, sustainability reports; *df*, degrees of freedom; *p*-Value, probability value or significance (two-tailed).

Verification of the statistical treatment assumptions: multicollinearity.

- Binary logistic regression

**Table A3.** Statistical results of the correlation between the independent variables.

| Research Variables | | Correlations Matrix | | | | |
|---|---|---|---|---|---|---|
| Independent | | Statistical Parameter | BV | AS | UM | SR |
| BV | Business volume | Pearson correlation | 1 | 0.196 | 0.000 | 0.168 |
| | | *p*-Value | - | 0.003 | 1.000 | 0.010 |
| | | N | 235 | 235 | 235 | 235 |
| AS | Activity sector | Pearson correlation | 0.196 | 1 | 0.059 | −0.233 |
| | | *p*-Value | 0.003 | - | 0.365 | 0.000 |
| | | N | 235 | 235 | 235 | 235 |
| UM | UNGC NP members | Pearson correlation | 0.000 | 0.059 | 1 | 0.059 |
| | | *p*-Value | 1.000 | 0.365 | - | 0.370 |
| | | N | 235 | 235 | 235 | 235 |
| SR | Sustainability reports | Pearson correlation | 0.168 | −0.233 | 0.059 | 1 |
| | | *p*-Value | 0.010 | 0.000 | 0.370 | - |
| | | N | 235 | 235 | 235 | 235 |

Note: BV, business volume; AS, activity sector; UMs, UNGC NP members; SR, sustainability reports; *N*, number; *p*-Value, probability value or significance (two-tailed).

**Table A4.** Statistical results of the collinearity coefficients of the research variables.

| Research Variables | | Collinearity Statistics | | |
|---|---|---|---|---|
| Independent | | Model | Tolerance | Variance Inflation Factor (VIF) |
| BV | Business volume | | 0.913 | 1.096 |
| AS | Activity sector | 1 | 0.883 | 1.133 |
| UM | UNGC NP members | | 0.990 | 1.010 |
| SR | Sustainability reports | | 0.893 | 1.120 |

Note: BV, business volume; AS, activity sector; UMs, UNGC NP members; SR, sustainability reports; VIF, variance inflation factor.

**Table A5.** Statistical results of the collinearity diagnostics of the research variables.

| Model | Dimension | Eigenvalue | Condition Index | Variance Proportions | | | | |
|---|---|---|---|---|---|---|---|---|
| | | | | **Constant** | **BV** | **AS** | **UM** | **SR** |
| | 1 | 3.029 | 1.000 | 0.020 | 0.040 | 0.030 | 0.030 | 0.030 |
| | 2 | 0.770 | 1.983 | 0.000 | 0.090 | 0.010 | 0.880 | 0.010 |
| 1 | 3 | 0.615 | 2.219 | 0.000 | 0.000 | 0.250 | 0.000 | 0.480 |
| | 4 | 0.419 | 2.690 | 0.070 | 0.870 | 0.110 | 0.090 | 0.060 |
| | 5 | 0.166 | 4.273 | 0.900 | 0.000 | 0.610 | 0.010 | 0.420 |

Note: BV, business volume; AS, activity sector; UMs, UNGC NP members; SR, sustainability reports.

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
