# Peer review of "The Reporting of SDGs by Quality, Environmental, and Occupational Health and Safety-Certified Organizations"

_sustainability, doi:10.3390/su11205797_

Round 1

Reviewer 1 Report

The paper aimed at mapping SDGs reporting among Portuguese organizations who had Quality, Environment, and occupational health and safety certificate.

the literature review was comprehensive enough and methodology was clear, results were clearly stated, although some adjustments could be added regarding the credibility and validity measurements of results (it is suggested that the authors write a few lines how they assured credibility and validity of their research findings).

The discussion section needs to be improved, it looked like conclusion, but the real discussion should compare the  finding of current research with the previous ones. stating that this results confirm the previous researchers findings or is in line with those researches is not clearly define the value added and contributions of this study, so the scientific contribution of study should be clearly stated in the abstract and also it should be discussed in a comprehensive way in the discussion section. one point could be to talk bout the differences, in which ways this study differs from the previous researches.

Author Response

Dear Reviewer,

Thanks for your valuable feedback and the suggestions to further improve our manuscript quality. We have endeavored to reply to your comments (and those of other Reviewers), that are highly appreciated. Please see the attached file with the revised paper version.

Kind Regards

The corresponding Author

C1. The literature review was comprehensive enough and methodology was clear, results were clearly stated.

Reply: Thanks.

C2. … although some adjustments could be added regarding the credibility and validity measurements of results (it is suggested that the authors write a few lines how they assured credibility and validity of their research findings).            

Reply: Please see sections 3.3. Materials and Methods and 4. Results, plus Appendix A.

C3. The discussion section needs to be improved, it looked like conclusion, but the real discussion should compare the finding of current research with the previous ones. stating that this results confirm the previous researchers findings or is in line with those researches is not clearly define the value added and contributions of this study, so the scientific contribution of study should be clearly stated in the abstract and also it should be discussed in a comprehensive way in the discussion section. One point could be to talk about the differences, in which ways this study differs from the previous researches.    

Reply: Please see section 5. Discussion and conclusions: “This research aims to map the present engagement level of those companies in addressing and reporting the SDGs”; “These results are consistent with Schramade [74] and Rosati and Faria [75] conclusions that most companies currently do not mention the SDGs in their reports. Although 23.8% is higher than the 16% found by Rosati and Faria [75], this indicates that there is still considerable room for improvement in this regard”;  … “Finally, the results also confirm that the organizations that publish sustainability reporting are more prominent in reporting SDGs, supporting Lozano [72] claims that sustainability reporting can be an essential driver of an organization's sustainability orientation”. … “OECD [96] specifically highlights the relevance of SMEs for  promoting the more economic SDGs, such as SDG 8 and SDG 9 in the OECD. This is supported by the results of this investigation that have identified SDG 09 - Industry, innovation, and infrastructure (21.3%), SDG 08 - Decent work and economic growth (20.0%) amongst the most reported SDGs by the Portuguese QEOHS certified companies.”

Reviewer 2 Report

Dear authors! The article is devoted to a relevant topic. A qualitative review of regulatory documents and scientific literature on the topic is presented. The study is based on a representative sample of Portuguese organizations.
BUT! What is the scientific purpose of the article? What scientific problem do you solve? Do you want to explore the focus of Portuguese organizations on the SDGs? Why should international readers be interested? The behavior of Portuguese companies in relation to the SDGs is typical for countries in Europe (World)?
The study has a lot of limitations: only Portuguese companies are considered, only the data of their websites is examined, resources and opportunities and sustainability indicators are not studied. You are missing out on the most interesting things now, making a foundation for the future .. It is necessary to explain why your results are useful to foreign scientists, politicians and managers.
There are no comments on the use of statistical tools for data processing.

Author Response

Dear Reviewer,

Thanks for your valuable feedback and the suggestions to further improve our manuscript quality. We have endeavored to reply to your comments (and those of other Reviewers), that are highly appreciated. Please see the attached file with the revised paper version.

Kind Regards

The corresponding Author

C1. Dear authors! The article is devoted to a relevant topic. A qualitative review of regulatory documents and scientific literature on the topic is presented. The study is based on a representative sample of Portuguese organizations. BUT! 

Reply: Thanks.

C2. What is the scientific purpose of the article? What scientific problem do you solve? Do you want to explore the focus of Portuguese organizations on the SDGs? Why should international readers be interested? The behavior of Portuguese companies in relation to the SDGs is typical for countries in Europe (World)?      Text was added to make these issues clearer.

Reply: Please see section Abstract: “Organizations can play a significant role in the advancement of Sustainable Development, and companies with Quality, Environmental and Occupational Health and Safety (QEOHS) certified management systems address the three Sustainability Dimensions (economic, environmental and social). This research aims to map the present level of engagement of those companies in contributing and reporting to the 17 Sustainable Development Goals of The United Nations 2030 Agenda, and therefore, support this Agenda implementation. … This study can be useful for both managers and decision-makers that aim to support organizations to contribute to the Sustainable Development Goals and achieving a better and sustainable future for all”.

Please see Section1. Introduction: “…Progress in implementing the United Nations 20230 Agenda should be assessed periodically by each country, involving governments, civil society, business and other stakeholders. In Portugal, the responsibility for overall SDG coordination rests with the Ministry of Foreign Affairs in liaison with the Ministry of Planning and Infrastructure, involving the other Ministries with their SDGs related tasks. The Inter-Ministerial Committee on Foreign Policy (CIPE) acts as the headquarters and forum for inter-ministerial coordination, both for the implementation of the SDGs and for the preparation of reports that will support national, regional and global monitoring processes.

Portugal is a European Union (EU) country, member of the Organization for Economic Co-operation and Development (OECD), and both Portugal (e.g. via EU and OECD) and the Portuguese (e.g., now, via the present UN Secretary General; in the past, via, the former EU Commission President) actively engages in international partnerships and institutions. It is expected, therefore, that this research can be replicated in other countries that also want to foster the UN 20230 Agenda, via the monitoring and implementation of the SDGs.   

This research aims to map the reporting of SDGs by Portuguese organizations that hold simultaneously Quality, Environmental and Occupational Health and Safety certifications. Since these International Standards already address (at least partially) the economic, environmental and social dimensions of SD, this investigation can contribute to gather further knowledge concerning SDGs adoption and foster its application by those organizations. By mapping the present level of engagement of those companies in contributing and reporting to the 17 Sustainable Development Goals of The United Nations 2030 Agenda, leading practices and areas for improvement can be identified, creating awareness and supporting decision and policy makers to further promote this Agenda implementation.  These companies can promote inclusive and sustainable economic growth, providing employment and decent work for all, advancing sustainable industrialization and fostering innovation, and reducing inequalities, by engaging in favor of the SDGs”.    

C3. The study has a lot of limitations: only Portuguese companies are considered, only the data of their websites is examined, resources and opportunities and sustainability indicators are not studied. You are missing out on the most interesting things now, making a foundation for the future .. It is necessary to explain why your results are useful to foreign scientists, politicians and managers.      

Reply: We agree with the Reviewer on the limitations of our research. Please see section 5. Discussion and conclusions that was further developed: “Although this research makes a novel contribution to the SDGs body of knowledge, particularly within the QEOHS certified organizations, it suffers from some limitations to be acknowledge when generalizing its findings. First, the sample is restricted to Portuguese organizations with certified QEOHOS management systems. Second, the investigation is restricted to the reporting of SDGs in company reports available on web sites without evaluating the performance in terms of SDGs advancement. Third, other organizational factors such as resources and capabilities and sustainability performance were not investigated. Fourth it should be assessed if this results and conclusions can be generalized to other EU and OECD countries, as there is research underlining statistical differences in environmental and social performance between developed EU countries and developing EU countries [97]. Moreover, countries must interpret the SDGs according to their national circumstances and levels of development [98]. Fifth, the SDGs logic imply that there is mutual dependence between SDGs (drawing analogies with integration of QEOHS Management System) [98], but this was not subject of this research.

Therefore, future research could be carried to evaluate the evolution of SDGs reporting with time, the relationships between SDGs, and consider other organizations than those QEOHS certified, and in other countries. Also, the application of data mining process, such as Cross Industry Standard Process for Data Mining (CRISP-DM), could be adopted allowing the application of text mining methodology on the digital archives of the sampled companies”. … This study contributes to the sustainable development and sustainability reporting body of knowledge by  mapping  the present engagement level of QEOES certified Portuguese companies in addressing and reporting the SDGs and it can be of value for other researchers that want to investigate and contribute to UN 20230 Agenda. It can be useful for decision and policy makers that aim to support organizations to contribute to the Sustainable Development Goals and the adoption of the UN 2020 Agenda. From a managerial perspective, it highlights that QEOHS certified organizations that already address the economic, environmental and social dimensions, due to its management systems certification can be more ambitious and match their strategies and actions with the relevant SDGs and report accordingly. Considering that the SDGs goals reporting is still modest, it would be desirable to have a more intense pressure from stakeholders to promote it, and have more prominent companies disseminating the adoption of SDGs within their supply chains, with the aim of achieving a better and sustainable future for all.

C4. There are no comments on the use of statistical tools for data processing.  

Reply: Please see sections 3.3. Materials and Methods and 4. Results, plus Appendix A: e.g.,  “ .. Data was analyzed dichotomously, assigning to the item the code or value “1 – one” (if present), otherwise, assigning to the item the code or value “0 – zero” (Haniffa and Cooke [90], p. 405). Software IBM SPSS Statistics® version 22 (International Business Machines – Statistical Package for the Social Sciences) and macro KALPHA version 2007 (macro Krippendorff’s α) were adopted to conduct the statistical calculations, hypotheses testing and reliability assessment”.  It was added in suggestion for future research “Also, the application of data mining process, such as Cross Industry Standard Process for Data Mining (CRISP-DM), could be adopted allowing the application of text mining methodology on the digital archives of the sampled companies “.

Reviewer 3 Report

I perceive this article as important and interesting. Issues of what and how is communicated by companies in the field of SDGs reporting require research, analysis and presentation of results.

Although the hypotheses seem to be quite obvious, research questions about what and how companies communicate SDGs let a reader know how the 17 Sustainable Development Goals of the United Nations 2030 Agenda are being achieved.
The only caution I have to the article is the lack of defining
the term 'communication' and discussing the relationship
between the terms 'communication' and 'promotion' in this particular research. The reader gets the impression that both of these terms are synonymous. In communication sciences,
these terms are clearly defined, and the reader need not be
familiar with such literature. Similarly, the relationship between ‘communication’ and ‘reporting’ should be explained for the sake of clarity.

Author Response

Dear Reviewers,

Thanks very much for the valuable feedback, which is highly appreciated.

C1. We reviewed the manuscript to try to clarify the concepts of "reporting," "communication," and "promotion." Sustainability reporting, in line with the literature, is defined in the manuscript as "the practice of reporting publicly on an organization's economic, environmental and social sustainability impacts and SDGs reporting as the practice of reporting publicly on how an organization addresses the SDGs. We have clarified that "The adoption of an internationally recognized framework, such as the SDGs, for sustainability reporting, and the report subsequent public disclosure to the relevant stakeholders (e.g., via their institutional websites), can provide a reinforced legitimacy to the organizations that pursue this approach." Please see the highlighted text for further details.

C2. We replaced the word promote by "encourage" or "advance" to avoid misunderstandings with the marketing and communication fields.

C3. A few editorial changes were made to differentiate this manuscript from previous research projects that preceded this investigation. Additionally, spaces between paragraphs were eliminated, and bullets were used to highlight the research hypotheses. Appendix A was arranged according to the journal template.

The revised paper version highlights the changes made to the original version.

Kind Regards

The corresponding Author

Reviewer 4 Report

Dear authors,

Please see attached my comments.

Author Response

Dear Reviewer,

Thanks for your valuable feedback and the suggestions to further improve our manuscript quality. We have endeavored to reply to your comments (and those of other Reviewers), that are highly appreciated. Please see the attached file with the revised paper version.

Kind Regards

The corresponding Author

C1. In general, the topic of this paper is interesting. The subject could be relevant and appropriate for the Sustainability. Here follow some points that need further attention.

Reply: Thanks.

C2. Introduction and background sections: Please, be more critical in addressing the research gap. What is the contribution of the paper to the literature? Emphasize these aspects already in the introduction to make paper attractive for readers.  

Reply: Thanks, text was added to make these issues clearer.

Please see section Abstract: “Organizations can play a significant role in the advancement of Sustainable Development, and companies with Quality, Environmental and Occupational Health and Safety (QEOHS) certified management systems address the three Sustainability Dimensions (economic, environmental and social). This research aims to map the present level of engagement of those companies in contributing and reporting to the 17 Sustainable Development Goals of The United Nations 2030 Agenda, and therefore, support this Agenda implementation. … This study can be useful for both managers and decision-makers that aim to support organizations to contribute to the Sustainable Development Goals and achieving a better and sustainable future for all”.

Please see Section1. Introduction: “…Progress in implementing the United Nations 20230 Agenda should be assessed periodically by each country, involving governments, civil society, business and other stakeholders. In Portugal, the responsibility for overall SDG coordination rests with the Ministry of Foreign Affairs in liaison with the Ministry of Planning and Infrastructure, involving the other Ministries with their SDGs related tasks. The Inter-Ministerial Committee on Foreign Policy (CIPE) acts as the headquarters and forum for inter-ministerial coordination, both for the implementation of the SDGs and for the preparation of reports that will support national, regional and global monitoring processes.

Portugal is a European Union (EU) country, member of the Organization for Economic Co-operation and Development (OECD), and both Portugal (e.g. via EU and OECD) and the Portuguese (e.g., now, via the present UN Secretary General; in the past, via, the former EU Commission President) actively engages in international partnerships and institutions. It is expected, therefore, that this research can be replicated in other countries that also want to foster the UN 20230 Agenda, via the monitoring and implementation of the SDGs.   

This research aims to map the reporting of SDGs by Portuguese organizations that hold simultaneously Quality, Environmental and Occupational Health and Safety certifications. Since these International Standards already address (at least partially) the economic, environmental and social dimensions of SD, this investigation can contribute to gather further knowledge concerning SDGs adoption and foster its application by those organizations. By mapping the present level of engagement of those companies in contributing and reporting to the 17 Sustainable Development Goals of The United Nations 2030 Agenda, leading practices and areas for improvement can be identified, creating awareness and supporting decision and policy makers to further promote this Agenda implementation.  These companies can promote inclusive and sustainable economic growth, providing employment and decent work for all, advancing sustainable industrialization and fostering innovation, and reducing inequalities, by engaging in favor of the SDGs”.    

C3. The authors should add a new Section – Literature review- where they should move Sections 1.1 – 1.3.        

Reply: Done.

C4. The authors could extend their Literature review with other interesting references related to the SDGs. Here are some suggestions: • Robert, K. W., Parris, T. M., & Leiserowitz, A. A. (2005). What is sustainable development? Goals, indicators, values, and practice. Environment: science and policy for sustainable development, 47(3), 8-21. • Assembly, G. (2015). sustainable Development goals. SDGs), Transforming our world: the, 2030. • Busu, M. Assessment of the Impact of Bioenergy on Sustainable Economic Development. Energies 2019, 12, 578. • Nilsson, M., Griggs, D., & Visbeck, M. (2016). Policy: map the interactions between Sustainable Development Goals. Nature News, 534(7607), 320.             

Reply: Thanks. These suggestions were considered and a few non-essential have articles been removed.

C5. Section 1.4 Research hypotheses should be moved to Section 2. Materials and Methods.    

Reply: Done.

C6. Section 3.1 should be moved to Materials and Methods. The sample description cannot be part of the Results section.

Reply: The sample description was moved to subsection 3.2. Data collection and sample in Section 3. Methodology. The descriptive analysis of the results was kept in section s 4.1. Descriptive analysis.

C7. The authors should relate the variables used in the Regression Model to the Literature. How is their model related to other models used in the economic literature and what is the novelty in their research.         

Reply: Please see text in section 3.3. Materials and Methods: “The proposed estimation model is supported by equation 2, which was based on the mathematical assumptions of binary logistic regression [91], [92]”.

C.8. The authors must give more information about the Sample used in their study. Why did they choose 235 organizations in their study? Is that enough? What sampling method did they use? Random? Stratified?             

Reply: Please see section 3.2. Data “In Portugal, the universe of the certified QEOHS organizations comprised, by 31 December 2017, a total of 698 organizations. The research sample (n) consists of 235 organizations, that is, all Portuguese organizations that were certified, within the scope of Quality (ISO 9001), Environment (ISO 14001), Safety and Health at Work (BS OHSAS 18001) as of December 31, 2017, and that made available an institutional website accessible on the Internet as of July 31 2019, where they release their institutional reports, at least one, concerning the last four (4) years”.

C9. In the Results section, the authors should provide and discuss an estimation of the parameters from equation (1). Are the they statistically significant? Is the model valid?               

Reply: Equation (1) mathematical formulation is in line with Carvalho et al. [76], Amran et al. [82], and Haniffa and Cooke [90]).

C10. Although it is not mandatory, the authors could add Conclusions section to discuss the main results and to add limitations and future research for their study.

Reply: Section Discussion was expanded and is now 5. Discussion and conclusions. The following text was added: “Portugal is a OECD member with a similar pattern as other OECD are countries, where small and medium enterprises (SMEs) are the predominant form of enterprise, accounting for approximately 99% of the business fabric, accounting for 70% of jobs and for creating between 50% to 60% of the added value [95]. According to OECD [96], SMEs have an important role to play in favor of the SDGs, as they can promote inclusive and sustainable economic growth, providing employment and decent work for all, promoting sustainable industrialization and fostering innovation, and reducing inequalities. OECD [96] specifically highlights the relevance of SMEs for  promoting the more economic SDGs, such as SDG 8 and SDG 9 in the OECD. This is supported by the results of this investigation that have identified SDG 09 - Industry, innovation, and infrastructure (21.3%), SDG 08 - Decent work and economic growth (20.0%) amongst the most reported SDGs by the Portuguese QEOHS certified companies. This research makes a novel contribution by  mapping the reporting of SDGs by Portuguese organizations that hold simultaneously QEOHS certifications. There is a considerable stream of research covering QEOHS certification and the organizations that adopt these International Standards already address (at least partially) the economic, environmental and social dimensions of SD. By mapping the present level of engagement of those companies in contributing and reporting to the 17 Sustainable Development Goals of The United Nations (UN) 2030 Agenda, leading practices and areas for improvement can be identified, creating awareness and supporting decision amongst both QEOHS certified organizations management, and policy makers, to further promote this Agenda implementation. We can conclude that Portuguese QEHOS certified companies have the potential to contribute to all the SDGs, and it is up to each company to identify which priority SDGs are based on their environmental, social and governance impacts along the value chain.

C11. The reference list must be reviewed. Please read the Author’s guidelines and Sustainability template and make the changes accordingly. - 94 reference is too much for a research paper. The authors should make the reference list shorter by keeping only the relevant references.            

Reply: Thanks. The reference list was reviewed. It is indeed a long list, because the paper addresses several research themes, such  as Quality, Environment, Occupational Health and Safety, ISO International Standards,  Management systems certification, Sustainable Development, Sustainability reporting and SDGs. Hence, the need to reference a considerable number of scientific works.

Round 2

Reviewer 4 Report

The paper is significantly improved.There are only some small editing issues to be solved:

Spaces between paragraphs are not necessary. Some empty lines must be deleted: 534, 545 etc. Appendix must be arranged according to the template.    Research Hypotheses could be placed in a new table or bullets could be used.

Author Response

(The authors gave the same response as above.)
